# The Use of Fecal Microbiota Transplant in Overcoming and Modulating Resistance to Anti-PD-1 Therapy in Patients with Skin Cancer

**DOI:** 10.3390/cancers16030499

**Published:** 2024-01-24

**Authors:** Tahne Vongsavath, Rodd Rahmani, Kyaw Min Tun, Vignan Manne

**Affiliations:** 1Department of Internal Medicine, University of Nevada, Las Vegas, NV 89154, USA; rodd.rahmani@unlv.edu (R.R.);; 2Department of Gastroenterology and Hepatology, University of Nevada, Las Vegas, NV 89154, USA

**Keywords:** melanoma, FMT, fecal microbiota transplant, dysbiosis, malignancy, anti-PD-1

## Abstract

**Simple Summary:**

While melanoma treatment has advanced and generally offers good results, treatment resistance remains a major source of morbidity and mortality in the patients it afflicts. While advances have been made in its treatment, there continues to be several patients that still have disease progression. The use of fecal microbiota transplantation has been used to augment the gut microbiome, decreasing overall inflammation, offering support in the treatment of other diseases such as *C. difficile* infection, with good utility. Thus, investigation of its use in other conditions and its ability to help augment medication effects is underway. This manuscript aims to review the use of FMT in advanced melanoma that has demonstrated treatment resistance.

**Abstract:**

While immune checkpoint inhibitors have evolved into the standard of care for advanced melanoma, 40–50% of melanoma cases progress while on therapies. The relationship between bacterium and carcinogenesis is well founded, such as in *H. pylori* in gastric cancers, and *Fusobacterium* in colorectal cancers. This interplay between dysbiosis and carcinogenesis questions whether changes in the microbiome could affect treatment. Thus, FMT may find utility in modifying the efficacy of anti-PD-1. This review aims to examine the use of FMT in treatment-resistant melanoma. A literature search was performed using the keywords “fecal microbiota transplant” and “skin cancer”. Studies were reviewed for inclusion criteria and quality and in the final stage, and three studies were included. Overall objective responses were reported in 65% of patients who were able to achieve CR, and 45% who achieved PR. Clinical benefit rate of combined CR/PR with stable disease greater or equal to 6 months was 75%. Reported objective responses found durable stable disease lasting 12 months. Overall survival was 7 months, and overall PRS was 3 months. As for the evaluation of safety, many patients reported grade 1–2 FMT related AE. Only following the administration of anti-PD-1 therapy were there a grade 3 or higher AE.

## 1. Introduction

Melanoma is a form of skin cancer with high incidence in both males and females, with rising incidence occurring particularly in White patients. Despite many advances in the treatment of the disease, it remains a major source of morbidity and mortality in the patients it afflicts. Most melanomas remain in the epidermis and if they are detected early enough are curable by simple surgical excision [1]. However, when melanoma becomes metastatic, a multifaceted approach becomes necessary to slow the disease. A cornerstone of modern treatment regimens involves identifying molecular alterations and implementing targeted gene therapies, most commonly involving the *BRAF* gene. In advanced disease or those without BRAF mutations, the use of immune checkpoint inhibitors (such as anti-programmed death 1 pathway, anti-PD-1) has been favored over more traditional regimens such as chemotherapy. Immune checkpoint modulators are generally involved with the maintenance of immunologic homeostasis, maintaining molecular self-tolerance [2]. Anti-PD-1 therapy blocks the immune regulatory pathway checkpoints that limit T-cell responses to melanoma and has clinically proven validity [3]. While there are continual advances in treatment, 40–50% of melanomas have been seen to progress even while on inhibitor therapies. This acquired resistance is generally associated with one or more features of poor prognosis such as stage M1c, elevated lactate dehydrogenase (LDH) level, or brain metastasis [4]. Prognostic factors following treatment failure also include LDH level, metastatic stage, progression site, tumor stage, and mutations status, each of which significantly correlate with survival [5]. Because treatment with anti-PD-1 therapy has become more relevant, there has also been an increase in organ specific immune-related adverse effects such as colitis, hypothyroidism, and possible pneumonitis especially when compared to chemotherapy or targeted drugs [6].

While the role of microbiota and cancer-related dysbiosis is not well studied in skin malignancies such as melanoma, its relationship elsewhere is well founded, such as in *H. pylori* in some gastric cancers, and *Fusobacterium* and *Streptococcus bovis* in colorectal cancers [7]. While theorized mechanisms include chronic inflammatory states that promote carcinogenesis via induction of proinflammatory toxins, and in alterations of signaling pathways, antibiotic treatment of these bacterial precursors allows alleviation of inflammation and dysbiosis, which further enhances the immune response. Furthermore, while cancer patients tend to have a basal amount of cancer-related dysbiosis, the use of chemotherapeutic agents tend to cause profound increase and further disrupt metabolic pathways. This inflammation tends to propagate further dysbiosis, creating an environment that cultivates bacterium that have better survivability in inflammatory states and thus further progress the cycle. In this inflammatory state, there is less production of bacterial derived short chain fatty acids, and further pro-inflammatory effects increase carcinogenesis [8]. Chronic inflammation as well as dysbiosis may reduce the efficacy of current treatments or medications used in symptomatic control during anti-PD-1 treatment by means of inhibiting absorption [9]. The use of antibiotics during immunotherapy have also been shown to negatively impact outcomes. Some forms of immunotherapy have been seen to cause disruption of the mucosal barrier causing subclinical colitis [10,11]. Dysbiosis itself, independent of other risk factors such as obesity has been shown to have carcinogenic effects [12]. This interplay between gut dysbiosis and carcinogenesis begs the question whether we could facilitate changes in the microbiome that could alter malignancy incidence rates, or treatment efficacy. Fecal microbiota transplantation (FMT) has been well studied in the treatment of refractory *Clostridiodes difficile* infection (*CDI*). The effect of the intestinal microbiome on several disorders, both intra- and extra-intestinal, as well as utility in medication absorption and efficacy are now under further investigation with promising early results [13]. While it has found utility in preventing the progression of neurologic disease as well as utility in obesity treatment, its longevity and utility are still under investigation [14,15]. Still, FMT remains standard of care for refractory or recurrent CDI as its utility in other conditions require further investigation [16]. Despite the lack of long-term safety data, FMT is widely accepted as generally safe without serious side effects [17].

In this review, we aim to further examine the current evidence for the use of FMT in treatment-resistant melanoma patients who have already or are going to receive anti-PD1 immunotherapy. Given the large percentage of melanoma patients who will experience resistance to immunotherapy, exploration of new strategies to overcome resistance are essential to improve patient outcomes. These studies may also play a role in improving our understanding of the connection between the gut microbiome and malignancy, and potentially add a positive potential benefit to a much broader patient population. As the intestinal microbiome helps to regulate immune function of the gut as well as augment the effect of immune modulators, we hope to examine the ability to manipulate the intestinal microbiota using FMT to modulate the efficacy of anti-PD-1 therapy in patients with advanced melanoma [18].

## 2. Materials and Methods

### 2.1. Search Strategy

We performed a comprehensive literature search across five databases (Pubmed/Medline, Embase, and Cochrane) using variations of the keywords “fecal microbiota transplant” and “skin cancer” to identify original studies published from inception through 15 July 2023. Results were limited to human studies that were available or published in English. There were a total of 121 studies available for review. See Appendix A for detailed search terms.

### 2.2. Eligibility Criteria

Inclusion criteria: (1) treatment with FMT by any delivery method or dose (2) patients with advanced metastatic melanoma also receiving anti-PD-1 therapy; (3) reporting of patient data and outcomes with FMT and anti-PD-1 therapy; (4) adult patients of any sex; (5) studies of at least moderate quality of evidence.

Exclusion criteria: (1) case reports which reflect unique cases and significant bias; (2) published abstracts, letters to editors, and commentaries which do not require detailed patient data or an extensive review process; (3) studies without patient data; (4) non-English studies; and (5) animal studies.

### 2.3. Quality Assessment

A series of quality assessment tools developed by the US National Heart Lung and Blood Institute (NHLBI) of National Institutes of Health (NIH) (https://www.nhlbi.nih.gov/health-topics/study-quality-assessment-tools, accessed 18 November 2023), were used to determine methodological quality and risk of bias for Before-After (Pre-Post) Studies With No Control Group. Similarly to NOS, a set of question items with Yes/No answers were used, with a “Yes” counting as a score of 1 and a “No” as a score of 0. In the tool used for Before-After (Pre-Post) Studies With No Control Group, there were a total of 12 questions. A score of 9–12 corresponds to good quality, while scores of 5–8 and 1–4 indicate moderate and poor quality, respectively [19,20].

In the final selection stage, only studies with at least a moderate level of evidence were included. Quality appraisal was performed by at least two of the following authors (T.V. and R.R). If there was any disagreement, a senior reviewer (K.T.) evaluated the article and achieved consensus through discussion. See the Appendix A for quality assessment scores for each study. See Appendix A for excluded studies.

The study selection process by preferred reporting items for systematic reviews and meta-analyses (PRISMA) is shown in the Appendix A, and was registered in the PROSPERO database [21]. The registered number is CRD42023445693.

### 2.4. Study Outcomes

The primary outcome for this study was to evaluate the efficacy of FMT to overcome or modulate resistance to anti-PD-1 therapy in its use for patients with advanced melanoma who have previously shown resistance to it or are naive to it. The reported response is variable and dependent on the study but is generally evaluated by the RECIST v1.1 criteria, which are a set of guidelines based on the original measurements set by the World Health Organization to classify lymphadenopathy grade [22].

The secondary outcome for this study is the safety of FMT use in this patient population, and the safety of combined FMT and anti-PD-1 use. Safety will be discussed using grading of adverse effects and the frequency of hospitalizations associated with each.

### 2.5. Study Selection and Data Extraction

A total of 122 articles were able to be retrieved on initial search. Two authors (T.V. and R.R.) independently reviewed these titles and abstracts, after which 11 articles were deemed relevant with patient data. Six of these texts were updates for three more recently published clinical trials and were condensed accordingly into three reviewable papers; one was a clinical protocol, and one was a case report. Full texts were then reviewed by at least two of the following authors (T.V. and R.R), and the updated trials were consolidated, after which three remaining studies fulfilled complete eligibility criteria. The case report described FMT use related to symptomatic treatment, and not for use of anti-PD-1 therapy and was excluded from data collection but briefly discussed. In case of disagreement, a senior reviewer (K.T.) arbitrated the final decision for inclusion. The study selection process by Preferred Reporting Items for Systematic Reviews and Meta-Analyses (PRISMA) statement is detailed in the Appendix A. Summary of included studies are shown in Table 1, while excluded articles are listed in the Appendix A. The IRB review was not required as all data were extracted from published literature and no patient intervention was directly performed.

## 3. Results

Through a literature search, we collected three articles describing the use of FMT for the modulation of anti-PD-1 therapy in advanced melanoma patients. In total these studies encompassed 45 adult individuals who had been diagnosed with advanced or metastatic melanoma [23,24,25]. A summary of the baseline characteristics for the included studies is provided in Table 2. While these studies evaluated the use of FMT in introduction or reintroduction of anti-PD-1 therapy, there were no instances in the included studies where FMT was used for refractory CDI or immune modulator medicated colitis. The type of intervention with FMT differed per each individual study in whether donor stool was from a healthy patient without history of melanoma, or patients with melanoma that either had or had not responded to anti-PD-1 therapy previously. The delivery route of FMT as well as dose and frequency were also unique per each author as Routy and Baruch delivered FMT prior to anti-PD-1 initiation while Davar administered it during anti-PD-1 treatment [23,24,25]. Routy, who treated his patients with FMT seven days prior to initiating the first cycle of anti-PD-1 with continuation of anti-PD-1 every 3–4 weeks, evaluated the efficacy of FMT in anti-PD-1 therapy in patients with advanced cutaneous melanoma who were anti-PD-1 naïve. Delivery was via a one-time oral capsule only [23]. Overall objective responses (OR), as determined by the RECIST v1.1 criteria, were reported in 65% of patients who were able to achieve complete response (CR), and another 45% who achieved partial response (PR). A table summarizing the reported efficacy of included studies is provided in Table 3. They did note that the clinical benefit rate of combined CR/PR with stable disease greater or equal to 6 months was 75%. Davar delivered single donor derived FMT that was given along with pembrolizumab, followed by pembrolizumab every 3 weeks until disease progression or intolerability. He reported objective responses using RESCIST v1.1 in three patients and durable stable disease lasting 12 months in three other patients. Overall survival was 7.0 months, and overall progression free survival (PFS) was three months. PFS in patients with disease control was 14.0 months [24]. Baruch who pretreated his patients with antibiotics for initial native microbiota depletion before colonoscopy delivered FMT at day 0 followed by oral ingested capsules at day 1 and day 12 and anti-PD-1 at day 14 repeating this regimen every 14 days for 6 cycles before going to anti-PD-1 monotherapy at day 90, reported clinical response in three patients, two PR and one CR, all of which had been treated by the same stool donor. All these responders crossed the 6-month progression free survival mark [25].

In a discussion of microbial change, Routy described findings at four different points in time: responder baseline compositions (S1), 1 week after FMT before anti-PD-1 therapy (S2), 1 month (S3), and 3 months (S4). He found that for both responder (R) and non-responder (NR) patients microbiomes moved toward their donors at S2, but NR patients regressed back towards their baseline at S3 and S4. R patients had similar microbiomes that further increased at S3 and S4 (*p* < 0.001 and *p* = 0.004). Furthermore, at S3, R patients had significant increased richness of *Ruminococcus* and *Faecalibacterium*, and decrease in *Clostridium methylpentosum*, *Enterocloster aldensis*, *Erysipelatoclostridum ramosum*, and *Enterocloster clostridioformis* [23]. Davar noted that while there was a higher alpha diversity in CR donors compared to PR donors at baseline, he found no significant difference in immunotherapy responses in patients who received stool from either donor. In R groups, he found that the phyla Firmicutes (Lachnospiraceae and Riminococcacae families) and Actinobacteria (Bifidobacteriaceae and Coriobacteriaceae families) were the most significantly enriched after FMT while the phylum Bacteroidetes decreased [24]. In Baruch’s study, patients were treated by two differing donors: donor 1 and donor 2. Of which, donor 2 was described to have a higher alpha diversity. After FMT treatment recipients of donor 2’s stool had a greater alpha diversity (*p* < 0.001) than those treated with donor 1 stool. While both donors showed favorable responses to immunotherapy both with high relative Lacnospiraceae, donor 1 showed high prevalence of Veillonellaceae and donor 2 showed high prevalence of Ruminoccaeae. Overall, posttreatment microbiomes showed increased immunotherapy favorable Veillonellaceae family and decrease in Bifidobacterium bifidum with donor 1 recipients showing higher *Bifidobacterium adolescentis* and donor 2 recipients showing higher *Ruminococcus bromii* [25].

As for evaluation of safety of FMT in anti-PD-1 therapy, Routy described 40% of patients which grade 1–2 FMT related AE occurred including diarrhea, flatulence, and abdominal discomfort. A summary of overall reported adverse events are included in Table 4. Following the administration of anti-PD-1 therapy were there a grade 3 or higher AE; including arthritis, fatigue, pneumonitis, and nephritis. Overall, 85% of patients reported any grade AE related to FMT or anti-PD-1 therapy or a combination of both with the majority reported within the first three months of anti-PD-1 initiation. He noted that overall, the addition of FMT did not increase the incidence of immunomodulator related AE [23]. While Davar reported overall mild AEs, each enrolled participant reported at least one, generally grade 1 AE. Three patients reported grade 3 AE including two fatigue episodes which resolved, and one hospitalization for peripheral motor neuropathy that was treated with intravenous immunoglobulin and corticosteroids and would eventually resolve [24]. Baruch reported only FMT related AE between days 3–15 described as mild bloating. Several grade 1 immune related AE were reported, and no grade 2–4 [25]. None of the authors reported infectious complication requiring antibiotics prior to FMT therapy or anti-PD-1 initiation and donors underwent extensive infectious screening.

## 4. Discussion

FMT is theorized to have utility in multiple disease processes due to its ability to augment the gut microbiome, it has been found to have an overall safe profile even in immunocompromised patients such as the ones studied here [26]. The use of FMT to augment the baseline microbiota is well established, with recipients’ baseline diversity and composition moving toward that of their donors with a general return to their original baseline. These studies suggest that responders maintained microbiomes similar to their donors after receiving FMT for longer periods of time than non-responders. Further, those with a baseline lower alpha-diversity, or the richness or diversity of species in a functional community such as the gastrointestinal tract, were also related to higher engraftment of donor species. As each individual’s microbiome is unique, there is no set definition of which composition defines a healthy gut, though is it generally accepted that stable and diverse microflora correlates with intestinal health [27]. So, these individuals with less alpha diversity may allow for a more receptive environment with less competition for resources. Interestingly, in Baruch’s study, while all FMT recipients and donors had high levels of similar taxa that had previously been associated with immunotherapy response, the only recipients who had CR had received stool from the same donor, donor one. The group of recipients who received stool from donor one was noted to have a relative abundance of taxa *Bifidobacterium*, which has been shown to be an effective modulator of immunotherapy [25]. In preclinical models by Sivan, different “signatures” of gut microbiota were described to exist. Favorable signatures were associated with enhanced intratumorally immune infiltration and enhanced systemic immunity allowing for greater response to treatment agents [28]. Several studies have also suggested that microbial response may depend more on the microbial diversity and richness of the donor’s stool rather than the recipient’s microbial environment, raising the question of the existence of “super donors” or individuals with significantly more successful FMT outcomes as stool providers [29]. It is possible that while donor 1 and donor 2 were both responders to anti-PD-1 treatment, that donor 1 had a basal microbiome that was more favorable for eliciting response to treatment. Although OR was noted in many patients with previously reported resistance, there may be question whether this could be a delayed response to previous anti-PD-1 treatment rather than an augmentation from FMT use. As suggested by Ribas, delayed response in metastatic melanoma patients who continued therapy beyond confirmed progression without influence of FMT was <8% making FMT driven augmentation more likely than delayed anti-PD-1 responses [30]. Baruch reported an abundance of *Bifidobacterium* and *Enterococcus* taxa, while Routy did not report a prevalence in any of these three taxa [23,25]. In an investigation of the microbial profiles of their patients, Routy utilized healthy stool donors with vastly different baseline microbiota and found there was no donor effect on outcomes evaluated, and that the microbiome of donors and recipients were similar after FMT and before anti-PD-1 therapy with the longest sustained effect in responder patients [23]. Baruch found that FMT use shifted microbiota composition toward favoring anti-PD-1 efficacy allowing previously anti-PD-1 patients with a previously unfavorable microbiome, to have a clinical response to anti-PD-1 [25]. When compared to Baruch, who reported higher engraftment rates, Routy and Davar showed no related correlation in outcome and rate of engraftment [23,24,25]. This may be related to the fact that Baruch’s participants underwent more than one session of FMT treatment, allowing for more colonization. Baruch also elected to pretreat his participants, allowing for better colonization of donor strains, possibly lending to his participants having more similar microbiomes to their donors after FMT [31]. Whether this affect can be described secondary to the correction of cancer-related dysbiosis allowing for a decreased inflammatory state or the promotion of regulated signaling pathways, FMT has demonstrated utility in the modulation of anti-PD-1 therapy in advanced melanoma. In studies completed on patients with renal cell carcinoma, it was demonstrated that while there were higher response rates when patients were receiving check point inhibitor therapy concomitantly with supplementation of live bacterial product, that the results were not significant [32]. This may offer an explanation as to why Routy’s group who received their FMT via one session of oral capsule only did not have a marked increase in any bacterial group, while the other two authors had delivery via endoscopy or endoscopy followed by 12 capsules. While studies have shown that generally oral capsule delivery of FMT is non-inferior to delivery via colonoscopy, there may be some limitations due to dose dependance or area of delivery [33].

The exact mechanisms of FMT augmentation in the use of anti-PD-1 and other immunomodulator therapies are still unknown. In these cases, we see an increase in bacterial families that have demonstrated immunotherapy favorable responses. In anti-PD-1 research in specific, *Bifidobacterium* has been noted to assist to potentiate anti-PD-1 monoclonal antibodies efficacy in melanoma mouse models, and its metabolite inhibited PD-1 expression activating natural killer immune cells helping to destroy tumor cells by perforin and interferon-gamma mediation [28,34]. Matson demonstrated the ability to manipulate the microbiome in germ free mice with specific taxa that enhanced therapeutic response with anti-PD-1 based therapy. Furthermore, their research evaluated patients before and after initial treatment and described specific species that were generally more abundant in first time responders, such as *Bifidobacterium longum*, *Collinsella aerofaciens*, and *Enterococcus faeceium* [35]. While generally accepted theories for the use of FMT in patients receiving immunotherapy treatment generally include the improvement of treatment related or cancer-related dysbiosis as well or the increase in immune supportive bacterium, the actual mechanism is likely a combination of these factors and then some.

While the overall safety of FMT is generally well proven, there was a notable increase in the grade of adverse effects when anti-PD-1 therapy was initiated. In a related case study by Groenewegen, two patients with advanced malignancy with refractory immune mediated enterocolitis (IMC) following treatment with anti-PD-1 therapy were treated with FMT [36]. Patient one, was treated with nivolumab and ipilimumab, leading to improved radiologic response. However, after the third treatment cycle he noted severe diarrhea. Biopsies of the colon showed severe pancolitis without CMV. He was treated with prednisone, five courses of infliximab, tacrolimus, and vedolizumab. These therapies failed to have a clinical response and was eventually treated with FMT. After an initial increase in stool looseness and frequency, he had gradual improvement onwards, and was noted to have an increase in *Collinsella*, *Bifidobacterium* and *Bacteriodies*. In a second patient with metastatic lung carcinoma who was receiving pembrolizumab combined with chemotherapy, treatment was stopped after three cycles because of endoscopically confirmed IMC with exclusion of infectious etiology and histological confirmation of diagnosis. He was treated with oral prednisone, two courses of infliximab, IV prednisone, tacrolimus, all without clinical improvement. The patient received three sessions of FMT in total. After the first session he had immediate improvement in the frequency of defecation however did have return of diarrheal frequency and was found to have *Campylobacter jejuni* infection. He was treated with meropenem, and feces sampled showed disturbed microbiology with low diversity and richness. Although he was treated, he continued to have increased diarrhea frequency and was treated with vedolizumab. There was no clinical response, and the patient was treated with two more sessions of FMT. Afterward frequency improved and stool bacterial richness and diversity reached that of the donor level [36]. While the use of FMT in these cases is not directly related to augmentation of anti-PD-1 therapy, they do describe the utility for aiding in the treatment of adverse effects associated with the therapy. It may be that pretreatment with FMT allows alleviation of baseline cancer-related dysbiosis not caused by anti-PD1 therapy and has the potential to offer some protective benefit from IMC, prior to the initiation of immune check point inhibitor therapy.

## 5. Limitations

An important limitation of this review was the small sample size of data amongst all relevant studies. There is scarce data on the use of FMT in patients receiving anti-PD-1 therapy and FMT. While these patients were all diagnosed with metastatic melanoma, there was no comparison arm for site of metastasis or progression of disease as well as lack of a control group. The type of anti-PD-1 therapy currently being used, and the types that were previously failed or were not controlled for and some patients had received prior anti-PD-1 therapy while others had not, possibly altering the microbiome to a significant amount. The studies reviewed also lacked an anti-PD1-only comparison group making absolute efficacy of FMT alone more difficult to ascertain. Finally, there were differences amongst studies in FMT donor characteristics, the use of antibiotic preparation, and FMT timing relative to anti-PD-1 initiation, preparation, and delivery. Without controlling for these factors, it is unclear if the use of antibiotics on the donor profile had significant effects on clinical response.

## 6. Conclusions

Our review highlights promising early findings in the use of FMT in the treatment of advanced melanoma refractory to immunotherapy and the safety of such treatment in these patients. While additional standardized trials are needed to prove the use of FMT in assisting to modify resistance to current immunotherapies, the initial data is promising. From the data available, it appears that the increase in richness and diversity of microbiota from FMT may assist in the modification of responses to anti-PD-1 therapy. While the exact mechanisms remain unknown, the increase in “immune favorable” bacterium and decrease in inflammation allows the body to mount a better respond to immunomodulator therapy. As in Baruch’s study, the evaluation of a “super donor” may be pertinent to an even more pronounced response with bacterial strains specific to improved response to anti-pd-1 therapy potentially aiding in future directions for treatment [25]. The optimal preparation and administration of FMT is unknown, its mechanism of effect is even less understood. Whether it acts to provide colonization of bacteria that outcompete pro-inflammatory bacteria, augments the metabolites that are being released, or assist with mucosal barrier inflammation leading to carcinogenesis, the mechanisms of this therapy’s potential widespread use outside of melanoma is worth further evaluation. While this therapy appears to be safe and effective in the very limited number of studies to date, larger clinical trials are needed to identify standardized treatment regimens, which specific patients may benefit from, and if the benefit will result in clinically significant changes in patient outcomes. As the study of FMT in the use of cancer therapy continues, there will likely be a shift towards individualized medicine. An individual’s microbiota has the potential to be a predictive biomarker in treatment response and may guide clinicians in treatment selection. Future research directed toward exploring the microbiomes modulation in medication efficacy and metabolism would allow for a more selective approach to treatment [34].

## Figures and Tables

**Table 1 cancers-16-00499-t001:** Summary of studies included.

Author/Year	Study Design	Received FMT N =	Included Population	Pre-FMT Treatment	FMT Intervention	Anti-PD-1 Intervention
Routy 2023 [23]	Phase 1, Single Arm Clinical Trial	20	Patients with confirmed unresectable or metastatic cutaneous melanoma with no previous anti-PD-1 treatment	None	Healthy donor stool delivered one time by oral capsules	1 week after FMT delivery (pembrolizumab 2 mg/kg every 3 weeks up to 2 years or nivolumab 240 mg every 2 weeks or 480 mg every 4 weeks ongoing)
Davar 2021 [24]	Phase II, two-phase Clinical Trial	15	melanoma patients who received at least two cycles of anti-PD-1 previously with non response	NR	Donor stool from advanced unresectable stage IIIB-D or metastatic melanoma treated with anti-PD-1 with ongoing complete or partial response in one cycle via endoscopy	one cycle of pembrolizumab within 3 days of fmt, followed by additional 2–4 cycles.
Baruch 2021 [25]	Phase 1, Clinical Trial	10	melanoma patients who had failed at least one line of anti-PD-1 therapy either as monotherapy or combination.	PO vancomycin 500 mg and neomycin every 6 h for 72 h	donor stool from metastatic melanoma patients who had underwent anti-PD-1 monotherapy and had CR for over 1 year delivered via colonoscopy and then 12 PO capsules	Day 14 after FMT, Nivolumab 3 mg/kg, q2 weeks for 6 cycles

**Table 2 cancers-16-00499-t002:** Summary of patient populations, malignancy type and tumor stage at study entry.

Study	Mean Age (yrs)	Male	Tumor Stage at Entry	Malignancy Type	Mutation Status	Prior Anti-PD-1 Treatment Failure (y/n)
Routy 2023 [23]	75.7 (48–90)	12 (60%)	unresectable stage III (2, 10%), M1a (3, 15%), M1b (9, 45%), M1c (3, 15%), M1d (3, 15%)	advanced cutaneous melanoma	BRAF (6, 30%), Non BRAF (14, 70%)	n
Davar 2021 [24]	61 (35–85)	11 (73.3%)	M1a (6, 40%), M1b (2, 13.3%), M1c (5, 33.3%), M1d (2, 13.3%)	refractory metastatic melanoma	BRAF (4, 26.7%), NRAS (3, 20.0%), wild type (8, 53.3%)	y
Baruch 2021 [25]	66 (49–69)	7 (70%)	M1a (3, 30%), M1b (1, 10%), M1c (4, 40%), M1d (2, 20%)	metastatic melanoma	BRAF (3, 30%), wild type (7, 70%)	y

**Table 3 cancers-16-00499-t003:** Summary of reported efficacy per study.

	Anti-PD-1 Used	Clinical Response	Objective Response	Complete Response	Partial Response	Progression Free Survival (mths)	Overall Survival (mths)
Routy 2023 [23]	Pembrolizumab or Nivolumab	NR	13 (65%)	4 (20%)	9 (45%)	NR	NR
Davar 2021 [24]	Pembrolizumab or Nivolumab	6 (40%)	3 (20%)	1 (6.67%)	2 (13.3%)	3	7
Baruch 2019 [25]	Nivolumab	3 (30%)	3 (30%)	1 (10%)	2 (20%)	NR	NR

**Table 4 cancers-16-00499-t004:** Overall reported AE per study. With (*) indicating the combined rate of AE due to FMT and anti-PD-1 therapy.

Author/Year	Grade 1–2	Grade 3–4	Grade 5, Hospitalizations
Routy 2023 [23]	40%, 60% *	24% *	0
Davar 2021 [24]	92.9% *	20% *	1 hospitalization *
Baruch 2021 [25]			
up to day 90	90% *	0	0
After day 90	80% *	0	0

## Data Availability

Data supporting the statements can be found on PubMed, Cochrane, and Google Scholar.

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
