# Peer review of "The Use of Fecal Microbiota Transplant in Overcoming and Modulating Resistance to Anti-PD-1 Therapy in Patients with Skin Cancer"

_cancers, 2024, doi:10.3390/cancers16030499_

Round 1

Reviewer 1 Report

Comments and Suggestions for Authors

The manuscript offers a comprehensive exploration of Fecal Microbiota Transplantation (FMT) in the context of treatment-resistant melanoma patients undergoing anti-PD1 immunotherapy. The authors meticulously outline their methodology for literature search and study selection, effectively summarizing the recent findings from pivotal studies, including Routy et al. 2023, Davar et al. 2021, and Baruch et al. 2021. This structured approach contributes significantly to the manuscript's strength. While the manuscript effectively presents key outcomes from these pivotal studies, there's an opportunity to bolster the discussion by delving deeper into microbiome analysis. Summarizing and deeply analyzing the microbiome dynamics alongside treatment responses in the aforementioned studies—Routy et al. 2023, Davar et al. 2021, and Baruch et al. 2021—would enrich the results and discussions section, and offer deeper insights into the interplay between the microbiome and treatment outcomes in melanoma patients.

Author Response

a section further describing the microbial results and relationships has been added per study.

Reviewer 2 Report

Comments and Suggestions for Authors

The article concerns possible immunity modification following fecal microbiota transplantation in skin malignancies treated by PD-1 inhibitors. The immune checkpoint inhibitors (ICI) are known to exert antitumor effects in some clinical entities. However, the success rates of ICI therapy show great inter-individual variability.

Intestinal microbiome may sufficiently modify antitumor response via immune surveillance mechanisms. The authors present a review of literature concerning safety and specific clinical effects of FMT in anti-PD-1-treated patients with advanced melanoma. The authors note absence of serious adverse effects following FMT. Concerning antitumor effects of FMT, only scarce information is presented from the selected sources due to minimal reliable data in the literature.

Remarks

Introduction: Line 57: An important prerequisite for FMT in melanoma patients is to alleviate treatment-related gut dysbiosis and to enhance antitumor immune response, rather than elimination of carcinogenic bacteria. Both viewpoints could be considered in the Introduction.

Line 60-70: The suggested mechanisms of bacteria-related carcinogenesis (early events) may be largely skipped, since the patients with advanced melanoma develop progressive tumor dissemination which needs systemic treatment.

Materials and methods:

Line 140: …grade of lymph nodes… better:  …lymphadenopathy grade…

Results.

Table 1 and Line 163: Of the 3 selected studies with FMT, one was performed prior to anti-PD1 therapy.  In two studies, FMT was administered to the patients who did not respond to ICI treatment. One study (Baruch et al.) concerned ICI-resistant cases treated with antibiotics just prior to FMT, thus expecting a sufficient dysbiosis of intestinal microbiota in this group.  Absent data on control (or comparison) groups in the articles should be also considered a limitation of this review.

In Table 1, one should note distinct clinical indications for FMT presented by the authors of the referred studies (associated infection?. ICI-related colitis?). Additional infectious complications prior to FMT (e.g., C.difficile infection) should be also mentioned and discussed if available (as the study of Baruch et al.).

Discussion:

Exact time intervals between conventional cytostatic/anti-infectious therapy and FMT should be also discussed.    

Line 218: … Bifidobacterium which has been shown to be immunotherapy favorable. – Maybe replaced by:  …an effective tool of immunotherapy?

Line 229: Please deciffer ‘OR’ abbreviation

Line 269: … overall safety of FMT is generally well tolerated… should be replaced by: … overall safety of FMT is generally well proven…

Some interesting clinical cases are discussed (e.g., line 270 and further) which concern clinical effects of FMT in PD-1-associated enterocolitis.

The conclusion, does not, however, contain any clear statements on presumed effects of FMT on the outcomes of ICI therapy in melanoma. Relative safety of FMT itself is, generally, a well proven fact shown in a number of studies.

Line-314-315: The statement on  … use of FMT appears to aid in overcoming resistance to current immunotherapies… should be formulated in more cautious manner.

   In general, the article is based in a limited number of studies with quite different clinical background and absence of clear FMT effects on the disease outcomes. The studies seem to lack the comparison groups (e.g., placebo).

Comments on the Quality of English Language

Extensive copy editing is required to avoid ambiquities in some sentences.

Author Response

Remarks

Introduction: Line 57: An important prerequisite for FMT in melanoma patients is to alleviate treatment-related gut dysbiosis and to enhance antitumor immune response, rather than elimination of carcinogenic bacteria. Both viewpoints could be considered in the Introduction.

added.

Line 60-70: The suggested mechanisms of bacteria-related carcinogenesis (early events) may be largely skipped, since the patients with advanced melanoma develop progressive tumor dissemination which needs systemic treatment.

partial section removed

Materials and methods:

Line 140: …grade of lymph nodes… better:  …lymphadenopathy grade…

agreed and addressed

Results.

Table 1 and Line 163: Of the 3 selected studies with FMT, one was performed prior to anti-PD1 therapy.  In two studies, FMT was administered to the patients who did not respond to ICI treatment. One study (Baruch et al.) concerned ICI-resistant cases treated with antibiotics just prior to FMT, thus expecting a sufficient dysbiosis of intestinal microbiota in this group.  Absent data on control (or comparison) groups in the articles should be also considered a limitation of this review.

Further description added to the limitations

In Table 1, one should note distinct clinical indications for FMT presented by the authors of the referred studies (associated infection?. ICI-related colitis?). Additional infectious complications prior to FMT (e.g., C.difficile infection) should be also mentioned and discussed if available (as the study of Baruch et al.).

the patients in the included studies did not have general indication for FMT such as refractory CDI or ICI colitis. Indication for treatment with FMT was in anticipation for anti-PD-1 therapy.

Discussion:

Exact time intervals between conventional cytostatic/anti-infectious therapy and FMT should be also discussed.   

timing addressed 

Line 218: … Bifidobacterium which has been shown to be immunotherapy favorable. – Maybe replaced by:  …an effective tool of immunotherapy?

changed

Line 229: Please deciffer ‘OR’ abbreviation

OR abbreviation key added

Line 269: … overall safety of FMT is generally well tolerated… should be replaced by: … overall safety of FMT is generally well proven…

Addressed 

Some interesting clinical cases are discussed (e.g., line 270 and further) which concern clinical effects of FMT in PD-1-associated enterocolitis.

The conclusion, does not, however, contain any clear statements on presumed effects of FMT on the outcomes of ICI therapy in melanoma. Relative safety of FMT itself is, generally, a well proven fact shown in a number of studies.

Line-314-315: The statement on  … use of FMT appears to aid in overcoming resistance to current immunotherapies… should be formulated in more cautious manner.

addressed 

   In general, the article is based in a limited number of studies with quite different clinical background and absence of clear FMT effects on the disease outcomes. The studies seem to lack the comparison groups (e.g., placebo).

agreed, addressed

Reviewer 3 Report

Comments and Suggestions for Authors

The authors presented the manuscript reviewing three articles describing the use of Fecal Microbiota Transplant in small groups of patients with metastatic melanoma, in 2 phase 1 and 1 phase 2 trials. In each study the different FMT intervention was performed so the results are hardly comparable. However, the application of FMT can decrease the inflammation what can improve the outcome of treatment. 

Chronic inflammation and dysbiosis may reduce the efficacy of current treatments or medications used during anti-PD-1-treatment by inhibiting absorption. So, if the FMT treatment could safely reduce the inflammation the immunotherapy would be more efficient for the patient.

The idea of the treatment of patients with gastro-intestinal problems with FMT is not new. However, the procedures for that are not fully worked out. The authors are very critical about that and described it in the limitations and conclusions. 

I have only doubts about the review based on three articles. May it should be published as an opinion.

Author Response

While we agree that the scarcity of data can be concerning, we have added or changed various wording instances to describe this possible effect of FMT on augmenting anti-pd-1 therapy with caution.

Round 2

Reviewer 2 Report

Comments and Suggestions for Authors

The authors have sufficiently revised the text in response to reviewers, however, without gross conceptual changes. Their literature analysis did not disclose specific mechanisms of FMT action in ICI-treated patients. Clinical heterogeneity and different timing of the reviewed trials is shown in more details (lines 175 … 193) being associated with heterogeneity of clinical outcomes observed by different research teams. Some additional limitations of study are specified, i.e., lack of a control group, differential FMT timing (lines 431, 348).

In Conclusion (line 354), the sentence …FMT appears promising in assisting in augmentation of resistance… may be replaced by a more realistic, for example: …Additional standardized trials on FMT are required in order to prove its modifying effects upon resistance to immunotherapies…

The manuscript still needs careful copy editing, especially in the added text fragments, to make the statements more clear and readable. E.g., Line 62: …. treatment for these bacterial precursors… is not clear. May be, the authors mean pathogenic bacterial strains, or metabolic precursors produced by these bacteria?

Line 355:  …in a significant of patients… : may be,  …in a significant subgroup of patients…? etc.

Comments on the Quality of English Language

Careful copy editing is required to provide better clarity of data analysis and  interpretation.

Author Response

The authors have sufficiently revised the text in response to reviewers, however, without gross conceptual changes. Their literature analysis did not disclose specific mechanisms of FMT action in ICI-treated patients. Clinical heterogeneity and different timing of the reviewed trials is shown in more details (lines 175 … 193) being associated with heterogeneity of clinical outcomes observed by different research teams. Some additional limitations of study are specified, i.e., lack of a control group, differential FMT timing (lines 431, 348).

The mechanisms in which FMT affects anti-pd-1 therapy is unknown, the theorized mechanisms are related to bacterium that are "immuno supportive" and related to correction of dysbiosis. A section has been added to the discussion and conclusion sections to address this.

In Conclusion (line 354), the sentence …FMT appears promising in assisting in augmentation of resistance… may be replaced by a more realistic, for example: …Additional standardized trials on FMT are required in order to prove its modifying effects upon resistance to immunotherapies…

Sentence has been addressed.

The manuscript still needs careful copy editing, especially in the added text fragments, to make the statements more clear and readable. E.g., Line 62: …. treatment for these bacterial precursors… is not clear. May be, the authors mean pathogenic bacterial strains, or metabolic precursors produced by these bacteria?

Line 355:  …in a significant of patients… : may be,  …in a significant subgroup of patients…? etc.

Copy and grammatical edits have been made.